# Nutritional, Bio-Functional, and Antioxidant Properties of Enzymatic Hydrolysates Derived from *Spirulina platensis* Proteins

**DOI:** 10.3390/foods14244242

**Published:** 2025-12-10

**Authors:** Ahmad Ali, Sanaullah Iqbal, Azmatullah Khan, Imtiaz Rabbani

**Affiliations:** 1Department of Food Science and Human Nutrition, University of Veterinary & Animal Sciences, Lahore 54000, Pakistan; ahmad.ali@uvas.edu.pk (A.A.); azmat.khan@uvas.edu.pk (A.K.); 2Department of Physiology, University of Veterinary & Animal Sciences, Lahore 54000, Pakistan; imtiaz.rabbani@uvas.edu.pk

**Keywords:** *Spirulina platensis* protein, protein hydrolysate, antioxidant activity, bioactive peptides, phytochemicals

## Abstract

*Spirulina (Arthrospira platensis)* is recognized as a high-protein microalga with potential for bioactive peptide production. In this study, *S. platensis* protein extract (~45% protein) was subjected to enzymatic hydrolysis using pepsin, trypsin, and chymotrypsin. A ~75% reduction in Bradford values indicated extensive protein breakdown, with degrees of hydrolysis of 15.6%, 21.4%, and 33.7% for pepsin-, trypsin-, and chymotrypsin-treated samples, respectively. SDS-PAGE confirmed the generation of low-molecular-weight peptides (<10 kDa). Hydrolysis caused only minor changes in amino acid composition, maintaining protein quality, with trypsin-hydrolysates showing the highest protein efficiency ratio (1.12) and biological value (78.83%). Antioxidant capacity increased significantly, with hydrolysates displaying a 33–68% rise in DPPH and 30–54% in FRAP activity, alongside a 33–44% reduction in lipid peroxidation. Furthermore, phytochemical content was markedly enhanced in hydrolysates compared to intact protein, with increases in total phenolic content (38–118%), total flavonoid content (59–78%), and terpenoids (24–37%). Among treatments, trypsin-SPPH (*Spirulina platensis* protein hydrolysate) consistently exhibited the most pronounced improvements. Collectively, these findings demonstrate that proteolysis of *S. platensis* proteins not only enhances antioxidant activity but also liberates bound phytochemicals, establishing *S. platensis* hydrolysates as promising functional food and nutraceutical ingredients.

## 1. Introduction

Plant-based foods are recognized as the principal source of essential nutrients in nature [1]. Microalgae, the primates of the kingdom Plantae, exhibit remarkable nutritional advantages over conventional plants, owing to their exceptionally high protein contents; balanced amino acid composition; and abundance of polyunsaturated fatty acids, vitamins, minerals, and bioactive compounds that collectively contribute to superior nutritional value and health-promoting potential [2]. Moreover, these microalgae also exhibit low costs in cultivation, with minimal water and land requirements, and high biomass production, making them more eco-efficient and a sustainable source of high-quality nutrients [3], with the potential to meet the demand of the rapidly growing global population. Apart from having a rich bioactive profile, these plant primates are gaining attention as safer alternatives to allelopathic treatments due to minimal adverse effects [4].

*Spirulina* (*Arthrospira platensis*), a filamentous cyanobacterium, has gained recognition in recent years due to its substantial protein content (55%) [5]. *S. platensis* exhibits high protein quality with 83–90% digestibility and a branched-chain amino acid-rich composition, making it more suitable for muscle protein synthesis and metabolic regulation. Protein quality indices of *S. platensis*, including the protein digestibility-corrected amino acid score (PDCAAS) and biological values (BVs), indicate a comparable protein quality with soy protein and casein [6].

To enhance its bioactive profile, *S. platensis* is further processed through hydrolysis, sonicated-assisted extraction, and fermentation [7]. With enhanced nutritional and functional properties, this microalga may serve as a regulator in multiple physiological processes. According to the literature, enzymatic hydrolysis is one of the most effective methods of obtaining bioactive peptides from *S. platensis* proteins [8]. This process employs proteases to cleave intact proteins, thereby releasing embedded bioactive peptides with nutritional and medicinal properties. In recent years, many studies have demonstrated the physiological implications of *Spirulina planensis* protein and its enzymatic hydrolysates, including antioxidant, antimicrobial, anticancer, anti-inflammatory, immunomodulatory, and hypo-cholesterolemic activities [9,10]. For instance, the in vitro antioxidant attributes of *S. platensis* protein and hydrolysates are reported in [11], with a significant reduction in DPPH and ABTS (up to 70%) [12], indicating more than four times more antioxidant capacity compared to the control [13]. On the other hand, in vivo studies are also available on zebrafish and mouse models, indicating improvements in SOD (26.8% increase) [14], catalase (45% increase) [15,16], and MDA (60% reduction) [14]. Otero and Verdasco-Martín [13] also observed the inhibition of pancreatic lipase by 30–31% and a reduction in serum triglycerides by 106% in an in vitro model. Conversely, in multiple in vivo mouse models, a 20% reduction in total cholesterol [17] and significant stimulation of neutrophils (40–42%) was observed [18]. Apart from these therapeutic effects, the anti-tumor effects of *S. platensis* proteins are also reported in [19], demonstrating cytotoxicity above 80% against colorectal adenocarcinoma, breast adenocarcinoma, prostatic adenocarcinoma cell lines, and in 4T1 cell-tumor-induced mice [20]. Additionally, protease-assisted hydrolysis has been increasingly applied to enhance phytochemical components by promoting the release of bound phenolics and modifying molecular structures to improve functional bioactivity, as evidenced by Wang, Wu, Liu, and Wu [21], indicating the use of enzymatic processing to extract valuable components, antioxidants, phenolics, etc., eventually aiding in increased bio-functional properties (i.e., antioxidant and antimicrobial) [22].

Oxidative stress, arising from excessive reactive oxygen species (ROS) production during mitochondrial respiration and disrupted redox homeostasis, contributes to cellular damage, apoptosis, and the progression of various non-communicable diseases (NCDs) [23]. Antioxidant defense mechanisms counteract these effects by scavenging free radicals and halting lipid peroxidation [24,25]. However, pharmacological interventions used to mitigate oxidative stress are often associated with multiple organ toxicities and cytotoxicity [26].

Consequently, naturally derived antioxidants, such as those found in *S. platensis*, have gained attention as safer and more biocompatible alternatives for maintaining cellular redox balance. *S. platensis* contains a wide range of antioxidant vitamins and minerals (vitamins C and E and zinc), serving as non-enzymatic defense against oxidative stress [27]. In addition to these, phycocyanin, phycobiliprotein, xanthophylls, and β-carotene pigments in *S. platensis* also display potent free radical scavenging capacity [28].

This nutritionally enriched and bio-functional profile of *S. platensis* has captured the attention of researchers, who have explored its health effects on various regimes, i.e., immunomodulation, antihypertension, and anti-diabetic [29]. However, the protein quality attributes and antioxidant potential of enzyme-digested hydrolysates remain insufficiently explored. Therefore, the present study aimed to establish the nutritional, bio-functional properties, and antioxidant potential of *S. platensis* protein-derived hydrolysates prepared using three different gastrointestinal proteases (pepsin, trypsin, and chymotrypsin). Furthermore, the comparative effectiveness of these two products was also assessed against raw *S. platensis* protein extract. The findings of this study can be used to redirect the use of *S. platensis* proteins and hydrolysates in functional food development.

## 2. Materials and Methods

### 2.1. Chemicals and Reagents

All chemicals used in this study were of analytical grade, purchased from Sigma-Aldrich (Steinheim, Germany), unless otherwise stated. *S. platensis* powder (SPP) was purchased from Scientific Trader-Pakistan. Pepsin (activity, 250 IU/mg, Lot-BCCD7625), trypsin (activity > 9000 IU/mg, Lot-SLCG5294), and chymotrypsin (activity, 800 IU/mg, Merck (Kenilworth, NJ, USA), Cat-no 804761) were used for proteolysis.

### 2.2. Proximate Analysis

Nutritional profiling of *S. platensis* whole powder was performed following the methods described in AOAC [30]. The moisture and dry matter contents (method no. 934.01), crude protein (no. 984.13), ether extract (no. 920.39), crude fiber (no. 978.10), and ash (no. 942.05) were experimentally estimated. Nitrogen-Free Extract (NFE), on the other hand, was calculated using the difference (no. 954.02) [30].

### 2.3. Formation of S. platensis Protein Extract (SPPE)

Using deionized water, a 5% solution of *S. platensis* whole powder (40 g SPP/800 mL DW) was prepared. After mixing thoroughly, the solution was subjected to freezing (−20 °C for 5 h) followed by thawing at room temperature (25 °C). After repeated freeze–thaw cycles (4 times), the mixture was treated with ice-bath ultrasonication (450 W for 30 min, 9 s interval every 6 s). Subsequently, temperature-controlled (4 °C) centrifugation was performed at 8694 RCF for 45 min. The supernatant with SPPE was obtained. The Bradford assay was performed to assess protein contents in SPPE and the extraction yield, using the method adopted from Fan et al. [31].

### 2.4. Enzyme Concentration Optimization and Formation of S. platensis Protein Hydrolysate (SPPH)

Enzymatic hydrolysis using pepsin, trypsin, and chymotrypsin was performed for SPPE (3%) under controlled conditions. To optimize hydrolysis, five different enzyme concentrations (IU) were evaluated to monitor the reduction in protein concentration relative to the increase in free amino acid yield. These concentrations were established by taking previous optimization experiments into consideration so that continuous hydrolysis efficiency could be ensured. For instance, an increase in trypsin enzyme concentration was reported to increase protein hydrolysis in pea protein isolates [32]. Similar optimization experiments are also documented in [33,34]. Therefore, for pepsin, enzyme concentrations ranging from 280 to 4500 IU, with stepwise 1-fold increments, were evaluated. Trypsin and chymotrypsin were tested at concentrations of 250–2000 IU and 125–1000 IU, respectively, following the same incremental approach. Hydrolysis with pepsin was conducted at pH 2 and 37 °C for 2 h, whereas hydrolysis with trypsin and chymotrypsin was carried out at pH 8 and 42 °C for 3 h. The hydrolysis conditions were selected from methods described in Wang et al. [35] and Fan et al. [31].

### 2.5. Degree of Hydrolysis

The degree of hydrolysis (DH) was determined following the instructions of [36] using the equation below.DH (%) = (10% TCA-soluble nitrogen in sample) ÷ total nitrogen in sample × 100

The TCA-soluble nitrogen method is a simple and reliable approach to determining DH, particularly for those hydrolysates in low-molecular-weight peptides that become soluble in 10%TCA solution.

### 2.6. SDS-PAGE

Stacking (4%) and resolving (12%) gels were used for polyacrylamide gel electrophoresis (PAGE). PAGE assembly was loaded with tris-glycine and 0.1% sodium dodecyl sulfate solution (8.3 pH). PAGE was conducted at 100 V until the dye reached the gel bottom. Subsequently, PAGE staining was performed using colloidal Coomassie blue, followed by destaining with 7% (*v*/*v*) acetic acid–water [37].

### 2.7. Solubility and Heat Stability

The guidelines of Nalinanon, Benjakul, Kishimura, and Shahidi [38] were used to calculate the solubility and heat stability of SPPE and SPPH:Solubility (%) = (protein contents in supernatant ÷ protein contents in sample) × 100Heat stability (%) = (protein contents in supernatant after heating ÷ protein contents in supernatant before heating) × 100

### 2.8. Nutritional Significance and Amino Acid Distribution

The amino acid profile of SPPE and SPPH was determined with a Biochrom 30+ Amino Acid Analyzer (Biochrom Ltd., Cambridge, UK), following the instructions of [39] with modifications. Moreover, the nutritional significance of SPPE and SPPH amino acid distribution was calculated using the equations below, focusing on amino acid score (AAS) [40], protein efficiency ratios (PER1-3) (Alsmeyer et al. [41]), and biological value (BV) (Morup et al. [42]).Amino acid score (AAS) = essential amino acids in sample ÷ essential amino acids recommended by FAO (%)PER 1= − 0.684 + 0.465 × Leu − 0.047 × ProPER2 = − 0.468 + 0.454 × Leu − 0.105 × TyrPER3= − 1.816 + 0.435 × Met + 0.78 × Leu + 0.211 × Hys − 0.944 × TyrBiological value (BV) = 102.15 × Lys^0.41^ × (Phe + Tyr)^0.6^ × (Met + Cys)^0.77^ × Thr^0.24^ × Trp^0.21^

### 2.9. Antioxidant Potential

The antioxidant activity of *S. platensis* protein extract and enzymatic hydrolysates was assessed using the 2,2-diphenyl-1-picrylhydrazyl (DPPH) radical scavenging assay and the ferric-reducing antioxidant power assay (FRAP) using guidelines described in Romulo et al. [43]. Lipid peroxidation assay was performed using TBARS, previously documented by De Leon et al. [44].

#### 2.9.1. Diphenyl-1-Picrylhydrazyl Assay (DPPH Assay)

Exactly 0.5 mL SPPE and SPPH samples were mixed vigorously with 0.1 mM DPPH solution and incubated for 20 min in a dark place. Decline in absorption was observed at 515 nm by a UV–visible spectrophotometer. Trolox (6-hydroxy-2,5,7,8-tetramethylchroman-2-carboxylic acid) dissolved in methanol (0–200 µM) served as the calibration standard, and antioxidant capacity was expressed as µmol Trolox equivalents (TE)/mL.Radical scavenging capacity (µmol Trolox Eq./mL) = (1 − Abs _Sample_ ÷ Abs _Blank_) × 100

#### 2.9.2. Ferric-Reducing Antioxidant Power Assay (FRAP)

The hydrolysate sample (0.5 mL) was mixed with a 2 mL FRAP solution and placed in a dark place (20 min). Decline in absorption was observed at 593 nm using a UV–visible spectrophotometer. A standard curve was prepared using 0–80 µM of FeSO_4_∙7H_2_O, and ferric-reducing capacity was calculated using the following equation.FRAP value (µM)= (Abs Sample ÷ Abs standard) × FRAP standard value (µM)

#### 2.9.3. Thiobarbituric Acid Reactive Substances (TBARS) Assay

Separate reaction tubes were prepared for SPPE and SPPH samples, along with a blank, after mixing 0.5 mL sample/blank, 0.5 mL phosphate buffer (pH 7.4), 0.1 M freshly prepared ascorbic acid, and 0.1 M ferric chloride. After mixing, incubation was performed at 37 °C (1 h) for lipid peroxidation. The reaction was stopped with the addition of 1 mL TCA (10%), and centrifugation was performed at 3000 RPM (10 min). Afterward, 1 mL of supernatant was added with 0.67% TBA (1 mL), followed by mixing. Heating was performed at 95 °C, after which ice cooling was performed. Absorption was taken at 532 nm using a UV–visible spectrophotometer. Lipid peroxidation was calculated using the following equation.MDA (µM/L): (Abs sample − Abs blank) × 6.41

### 2.10. Qualitative and Quantitative Phytochemical Analyses

The standard protocols used to perform qualitative phytochemical screening were selected from Shrestha et al. [45]. SPPE and SPPH samples were subjected to flavonoid, terpenoid, and total phenolic quantification following the guidelines described in Kalita et al. [46], Khanal et al. [47], and Al Jadidi et al. [48], respectively.

#### 2.10.1. Total Flavonoid Content (TFC)

A standards curve was prepared using 0.2–0.8 mg/mL catechin prepared in methanol. A sample of around 0.25 mL was taken with 1.25 mL DW, followed by mixing of 75 µL sodium nitrite (5%). After a 6 min stay, 150 µL aluminum chloride (10%) was added, and the volume was increased to 2.5 mL with DW. Absorption was taken at 510 nm, and TFC was expressed as the µg/mL catechin equivalent.

#### 2.10.2. Terpenoids

After taking a 5 mL sample in a Petri dish, around 10 mL of petroleum ether was added. The ether layer was separated using a funnel after 10 min, followed by drying. Terpenoids were calculated using the equation mentioned below.Total terpenoid content = (initial weight of sample − final weight of sample) × 100 ÷initial weight of sample (Wi)

#### 2.10.3. Total Phenolic Content (TPC)

The TPC was quantified using the Folin–Ciocalteu method based on redox reaction. A standard curve was prepared using 0–100 µg/g gallic acid. A sample of around 125 µL was mixed with 500 µL DW, followed by the addition of 125 µL FC reagent. After a 6 min stay, 1.25 mL Na_2_CO_3_ (7%) was added. The final volume was increased to 3 mL using DW. After 90 min of dark incubation, absorption was measured at 725 nm. The TPC was expressed as the µg/mL gallic acid equivalent.

### 2.11. Statistical Analysis

All the experiments of this study were conducted independently three times, with each analysis being conducted in triplicate, ensuring reproducibility and suitability for statistical analysis. Data were analyzed using R statistical software version 4.5.1 and expressed as mean ± SD (*n* = 3) and percentages. Data normality was assessed through the Shapiro–Wilk and Kolmogorov–Smirnov tests, with a *p*-value < 0.05 considered normally distributed. The differences in heat stability, antioxidant potential, and quantitative phytochemical analysis for SPPE and pepsin-SPPH, trypsin-SPPH, and chymotrypsin-SPPH were tested using one-way ANOVA. The intergroup differences were measured using the Duncan multiple range test (DMRt). The level of significance was kept at 5%.

## 3. Results and Discussion

### 3.1. Proximate Analysis of S. platensis (Whole Powder)

The 100 g *S. platensis* crude powder, when subjected to proximate analysis, contained 63.56% protein, 13.5% nitrogen-free extract (NFE), 7.09% ash content, 5.5% crude fat, 5.03% fiber, and 3.3% moisture content. The nutritional composition of *S. platensis* corresponds to most of the available literature, with compositional values residing in a similar range with minute variations [49,50]. Variation in composition may be attributed to nutrients in growth media, cultivation time, and conditions [51].

### 3.2. Protein Extraction and Yield

The freeze–thaw cycles and ultrasonic-assisted extraction yielded 452 mg/g of proteins, which corresponds to 45.2% protein yield in comparison to whole powder. This represents a reduction compared to 63.56% protein in dry biomass. The proteins in 5% (*w*/*v*) SPPE and whole *S. platensis* powder correspond to 22.63 mg/mL and 31.78 mg/mL, respectively. Comparable extraction yields have been reported in [31,52], with extraction efficiencies of 40.7% and 50%. The observed reduction can be the result of hydrophobic protein fraction losses and recovery method conditions (extraction time, solvent composition, temperature, etc.) [52]. Despite a reduction in proteins, the employed method resulted in high quantities of available proteins.

### 3.3. Enzyme Concentration Optimization for S. platensis Protein Hydrolysis

Final enzyme concentrations for the formation of *S. platensis* protein hydrolysate (SPPH) from proteolytic enzymes (pepsin, trypsin, and chymotrypsin) were selected based on optimization, as shown in Table 1. The Bradford assay demonstrates the protein concentration, and the ninhydrin test indicates free amino acid concentration in the hydrolysate. As enzyme conc. increases, a gradual decrease in protein concentration and a corresponding increase in the amino acid concentration can be observed since more free amino acids are available after hydrolysis at high enzyme concentration. The final enzyme concentrations of the employed gastrointestinal proteases were based on the optimal concentration displaying the highest bond cleavage. Pepsin displayed the highest protein hydrolysis at 4500 IU, as evidenced by the lowest Bradford (4.2 ± 0.05 mg/mL) and the highest ninhydrin (7.79 ± 0.05 mg/mL) values. Trypsin yielded similar results at 2000 IU. The chymotrypsin worked best at 1000 IU, with 7 ± 0.18 mg/mL Bradford and 6.91 ± 0.08 mg/mL ninhydrin values. The majority of *S. platensis* proteins were converted into smaller peptides after the pepsin/trypsin and chymotrypsin treatments, as a 70–80% decline was observed in the Bradford value. Fan, Cui, Zhang, and Zhang [31] also reported around 92–95% conversion of high-molecular-weight proteins (>10 KDa) into smaller peptides (1 KDa, 3–5 KDa, and <10 KDa).

Close observation of the digestive machinery reveals that proteins are cleaved by a variety of enzymes, whereas carbohydrates and fats are primarily digested through the action of amylase and lipase. The real reason lies in its composition, as proteins are composed of 20 different subunits known as amino acids. Proteolytic enzymes can only target specific cleavage sites. Pepsin primarily cleaves the peptide linkages around hydrophobic and aromatic residues in an acidic environment [53]. Chymotrypsin’s cleavage sites include tryptophan, tyrosine, and phenylalanine, with the ability to release hydrophobic peptides with possible bioactive capabilities. On the other hand, trypsin acts on the C-terminal of lysine and arginine (basic amino acids) with the potential to produce smaller and charged peptides. Moreover, lysin and arginine residues are frequently located on the bioactive regions of *S. platensis* proteins [54]. The superior bioactivity of trypsin-SPPH can be attributed to its charged peptides, capable of neutralizing free radicals. The heterogeneity of these enzymes with respect to cleavage sites, efficient hydrolysis, and specificity makes them useful for in vitro protein hydrolysis with the potential to generate bioactive peptides with a favorable size distribution [55].

### 3.4. Degree of Hydrolysis

In the present study, the degree of hydrolysis (DH) values of SPPE treated with pepsin, trypsin, and chymotrypsin were observed as 15.6%, 21.4%, and 33.7% respectively. On the other hand, Zhang and Zhang [56] reported DH values of 7.1% and 38.5% for pepsin and trypsin, respectively. Variations in DH can be attributed to the respective enzyme activity [57], processing time, nature of the protein being treated, and hydrolysis conditions [58]. Similarly, a collective DH of 31.49% was reported for these three enzymes (pepsin, trypsin, and chymotrypsin) in [35].

### 3.5. SDS-PAGE of SPPE and SPPH

SDS-PAGE results for *S. platensis* protein extract and its enzymatic hydrolysates (prepared from pepsin, trypsin, and chymotrypsin) are displayed in Figure 1. The electrophoresis pattern of SPPE revealed the presence of strong bands around 17–18 KDa and 55–60 KDa, corresponding to predominant water-soluble proteins and fractions of phycocyanin subunits. Additionally, faint to medium bands were also observed near 25, 37, and 150 KDa, indicating moderate- to high-molecular-weight proteins. Following hydrolysis, protein bands above 20 KDa were remarkably reduced, and light to weak bands were observed below 10–15 KDa across all types of hydrolysates. This reflects an effective and significant breakdown of intact protein in SPPE into its low-molecular-weight peptides, validating high proteolytic efficiency under optimized conditions. Comparable findings were observed in Fan, Cui, Zhang, and Zhang [31], who reported around 90% conversion of *S. platensis* proteins into a less-than-10 KDa fraction after pepsin and trypsin hydrolysis. Similarly, ref. [59] also reported distinct low-molecular-weight bands after pepsin and pancreatin hydrolysis. The observed degradation not only validates the effective digestion of proteins but also supports the formation of low-molecular-weight bioactive peptides with functional and nutraceutical applications.

### 3.6. Amino Acid Composition and Nutritional Significance

Amino acid composition and nutritional quality indices for SPPE and pepsin-, trypsin-, and chymotrypsin-based SPPH are demonstrated in Table 2. SPPE, along with its hydrolysates, was observed with the whole amino acid composition, mainly rich in glutamic acid, aspartic acid, leucine, and phenylalanine/tyrosine. This composition is consistent with studies reporting high protein quality in *S. platensis* [60,61].

Hydrolysis, on the other hand, has only resulted in a minor alteration in amino acid values, indicating that digestion of the protein did not result in reduced protein quality. Leucine remained relatively stable across all the hydrolysis treatments (3.88–4.03 g/100 g), while lysine showed a tiny reduction in chymotrypsin hydrolysates (2.76 g/100 g). The authors of [59] also reported no considerable changes in the amino acid distribution of *S. platensis* protein and its pepsin and pancreatin-based hydrolysates.

Considering protein quality parameters based on protein efficiency ratio (PER), biological value (BV), and amino acid score, SPPE was found to be superior compared to its hydrolysates, securing around 69% AAS and 74.25% BV. Among all hydrolysates, trypsin-SPPH was observed to have the highest level of PER1 (1.12) and BV (78.83%) compared to other hydrolysates (pepsin/chymotrypsin). Nevertheless, all SPPH and SPPE samples retained AAS (45%), fulfilling the FAO’s complete protein criterion [40]. Overall, SPPE and SPPH demonstrate comparable protein quality and amino acid scores, with trypsin-SPPH retaining most essential amino acids, indicating its bio-functionality and nutritious value.

### 3.7. Solubility and Heat Stability of SPPE and SPPH

Solubility is one of the most significant functional properties of protein because of its effect on other functions (foaming, emulsification, and gelation) and sensory properties (texture, color, etc.). Fundamentally, the solubility is based on protein–solvent and protein–protein interactions, as well as environmental conditions, including the pH, molecular weight, and polarity of amino acid groups present [62]. The solubility of the *S. platensis* protein extract and enzymatic hydrolysates was investigated at a pH ranging from 2 to 9, as displayed in Figure 2.

SPPE displayed a lower solubility at a pH ranging from 2 to 9 in comparison to its pepsin, trypsin, and chymotrypsin hydrolysates individually. As per conjecture, the solubility was highly pH-dependent since the acidic conditions facilitate the production of low-molecular-weight peptides, which significantly increase the number of ionizable functional groups in the sample [63,64].

Initially, the solubility was lower from pH 2 to 4 due to its proximity with the isoelectric pH (pH 4; zeta potential, ±0 mV), the pH at which the positive and negative charges on the molecular surface become equal, and hydrophobic interaction among proteins was much higher than hydrophilic and repulsive force interactions produced by charged residues [65]. These results are in complete accordance with the findings of Grossmann et al., who investigated the algal proteins from *Phaeodactylum tricornutum* and *Nannochloropsis oceanica* and observed the low solubility of these algae at pH 2 [66]. Above a pH of 4, the solubility of all the fractions gradually increases, with pepsin hydrolysate showing the greatest solubility, as compared to SPPE and other hydrolysates. At pH 4, SPPE had 36% solubility, while the pepsin hydrolysates had 46% solubility, which both increased to 83% and 93% at pH 9, respectively. The solubility of trypsin and chymotrypsin hydrolysates lies between this range. These results were highly consistent with Bispo et. al., who observed a 36% increase in solubility in *S. platensis* at pH 3 and 75% at pH 7 [62].

Thermal stability is the property that gives insight into the structural stability and bioactivity of proteins. All three SPPHs displayed higher stability than SPPE. SPPH-pepsin showed the highest (99.26%) stability, followed by SPPH-chymotrypsin (95.66%) and SPPH-trypsin (95.36%). A statistically significant difference was observed (*p*-value < 0.05) in the heat stability of SPPE and P-SPPH/T-SPPH/C-SPPH. Enzymatic hydrolysates appear to be more heat stable than SPPE, as protein is intact in SPPH, while in hydrolysates, the enzyme has already converted the protein into shorter peptide fractions. These results corroborate the previous findings by [67], where enzyme-treated *S. platensis* powder displayed no thermal peak because the enzyme had already denatured the protein into peptides [67].

Due to their aggregation ability under heat, intact proteins are considered heat-sensitive [38]. In contrast, hydrolysates (with low-molecular-weight peptides) are often heat-stable. This heat stability can be attributed to the loss of their secondary structure after hydrolysis [68], anti-aggregation ability secondary to reduced exposure to hydrophobic groups [69], and the presence of similarly charged side chains, inhibiting secondary structure formation [70].

### 3.8. Antioxidant Potential of SPPE and SPPH

DPPH, iron chelating activity, and lipid peroxidation assay results for SPPE and its enzymatic hydrolysates are displayed in Table 3. The trypsin hydrolysate demonstrated the highest antioxidant potential, with the highest IC50 value (0.0283 mg/mL), followed by SPPH-pepsin (0.0181 mg/mL) and SPPH-chymotrypsin (0.0119 mg/mL). The radical-scavenging activity was significantly higher among the enzymatic hydrolysates as compared to SPPE. A similar trend was observed for iron-chelating activity, as SPPH-trypsin demonstrated the highest antioxidant activity with a FRAP value of 0.0209 mg/mg. SPPH-chymotrypsin and SPPH-pepsin (0.0164 and 0.0138 mg/mlg^−1^, respectively) displayed comparatively less antioxidant potential than SPPH-trypsin, but they were higher than SPPE. On the flipside, the highest MDA levels were observed in SPPE compared to hydrolysates, indicating more lipid peroxidation in intact proteins compared to digested ones.

In our study, SPPH demonstrated significantly higher antioxidant value in comparison to intact SPPE across the DPPH and FRAP assays. Three types of hydrolysates showed a 33–68% increase in DPPH and a 30–54% increase in the FRAP value compared to intact protein, with trypsin-SPPH exhibiting the strongest activity among all treatments. Masoumifeshani, Abedian Kenari, Sottorff, Crüsemann, and Amiri Moghaddam [71] also observed around 78% DPPH scavenging and 74% FRAP activity in <3 KDa *S. platensis* protein peptides, outperforming intact proteins [71]. Similarly, Ma, Zeng, Zhou, Cheng, and Ren [72] demonstrated enhanced free radical scavenging and the reducing power of *S. platensis* hydrolysates compared to native protein [72]. It can be stipulated that the increase in free radical scavenging is due to the abundance of low KDa peptides in *S. platensis* hydrolysates.

Further halting capabilities of *S. platensis* hydrolysates were validated using the lipid peroxidation assay. A reduction of around 33–44% was observed in hydrolysates compared to intact protein in the present study, indicating stronger protection against oxidative membrane damage. Ma, Zeng, Zhou, Cheng, and Ren [72] reported a similar trend of protection in H2O2-stressed liver cells against lipid peroxidation and ROS generation. Enzymatic hydrolysis, with liberation of hydrophobic/aromatic amino acids (tyrosine, tryptophan, and phenylalanine), also validates the functional significance of *S. platensis* hydrolysates, as these released molecules are known to increase antioxidant properties.

The extensive proteolytic degradation evidenced by SDS-PAGE was further reflected in the antioxidant assays, indicating that the release of low-molecular-weight peptides contributed substantially to the radical-scavenging potential of *S. platensis* protein hydrolysates. It is well established that enzymatic hydrolysis enhances the antioxidant potential of proteins by exposing reactive amino acid residues and generating peptides with specific structural motifs capable of electron donation or hydrogen atom transfer [73]. The observed increase in antioxidant activity in pepsin-, trypsin-, and chymotrypsin-derived hydrolysates can, therefore, be attributed to the formation of these bioactive peptides, as supported by the molecular profiles obtained in SDS-PAGE. Comparable findings were reported in enzymatic hydrolysates of *S. platensis* and other microalgal proteins, where enhanced antioxidant capacity corresponded to a higher degree of hydrolysis and smaller peptide fractions [59,74]. These results collectively indicate a strong link between peptide size distribution and antioxidant efficiency, highlighting the functional relevance of enzymatic hydrolysis in improving the bioactivity of *S. platensis* proteins.

### 3.9. Phytochemical Analysis of SPPE and SPPH

Phytochemicals are non-nutritive functional components of food capable of bioactivity beyond basic nutrition and are generally considered the secondary metabolites of plants. Table 4 and Table 5 present qualitative and quantitative analyses of eight different phytochemicals in SPPE and pepsin-, trypsin-, and chymotrypsin-based SPPH. Alkaloids, flavonoids, saponins, terpenoids, and polyphenols were found among all types of hydrolysates and protein extracts. Glycosides were detected in all groups except SPPH-pepsin. Tannins were found to be present in SPPE and SPPH-pepsin, while they remained undetected in trypsin and chymotrypsin hydrolysates. Furthermore, phytosterol was absent in all groups.

Significant differences were observed in TPC, TFC, and terpenoids among all groups (*p*-value < 0.05). Meanwhile, SPPH-trypsin demonstrated the highest phenolic content (458.52 µg/mL), followed by SPPH-pepsin (290.76 µg/mL), in comparison to SPPE. The fewest phenolic contents were found in SPPE (211.28 µg/mL). A similar trend was observed in flavonoids of SPPE and multiple SPPH groups. The highest number of flavonoids was observed in trypsin-based SPPH (3.98 mg/mL), followed by pepsin (3.55 mg/mL). Moreover, raw SPPE was found to have higher TFC (2.24 mg/mL) in comparison to chymotrypsin-SPPH (2.12 mg/mL). Terpenoids displayed a slightly different trend. Around 10%, 9.4%, 8.6%, and 7.63% terpenoids were found in pepsin-SPPH, trypsin-SPPH, chymotrypsin-SPPH, and SPPE, respectively.

The impact of hydrolysis on total phenolics has been studied in various food products, including rice bran and quinoa, which reported an increase of 75 to 144% TPC, respectively, after hydrolysis [75,76]. To the best of our knowledge, no study has directly compared the phytochemical quantification of *S. platensis* hydrolysates and intact protein. An increase in TPC (38–118%), TFC (59–78%), and terpenoids (24–37%) in *S. platensis* hydrolysate in the present study confirms that proteolytic cleavage has indeed facilitated the release of these bound phytochemicals. This suggests that the enhanced antioxidant activity observed in *S. platensis* hydrolysates is not only due to bioactive peptides but also to the greater availability of phytochemicals liberated through enzymatic hydrolysis.

*S. platensis* (raw powder), along with its derived extracts and hydrolysates, is rich in phytochemicals, i.e., polyphenols, flavonoids, and carotenoids, indicating its potent antioxidant potential. Polyphenols neutralize the ROS by donating electrons and hydrogen atoms, acting as free radical scavengers and by stabilizing phenoxyl radicals, through which these bioactive compounds can terminate the cascading oxidative reactions [77]. Flavonoids, another major bioactive component of *S. platensis*, can exhibit free radical scavenging activity, as well as metal ion chelation. The presence of ortho-dihydroxyl (catechol) in the B-ring of flavonoids can prevent metal-catalyzed free hydroxyl radicals [78]. Moreover, microalgal-related polyphenols and flavonoids are also linked with the upregulation of SOD, catalase, and GPx, strengthening the cellular redox defense in in vivo models [79]. These combined effects of direct free radical neutralization explain the antioxidant activity of *S. platensis* protein extract and its derived hydrolysates.

The present study was limited to the use of crude protein extract and its derived hydrolysates rather than purified protein fractions. Despite optimization of the hydrolysis conditions to evaluate the stepwise degradation of intact proteins, the extract and its hydrolysates likely retained certain bioactive constituents, such as phenolics, flavonoids, and other pigmented molecules. These co-existing compounds could have increased or interfered with the measured antioxidant properties of *S. platensis* proteins and derived hydrolysates. Therefore, the observed beneficial effects cannot be solely attributable to the *S. platensis* protein and hydrolysates. Future studies employing the purified hydrolysates and peptide fractions are warranted to better delineate the specific contributions of these *S. platensis*-derived peptides regarding observed bioactive properties.

## 4. Conclusions

This study demonstrated that enzymatic hydrolysis of *S. platensis* proteins using pepsin, trypsin, and chymotrypsin produced low-molecular-weight peptides with superior antioxidant activity and stronger lipid peroxidation inhibition than the native protein. Enhanced levels of phenolics, flavonoids, and terpenoids indicated that proteolysis released bound bioactive compounds, improving functional potential. The main innovation lies in linking enzyme-specific hydrolysis to structural and phytochemical modifications in *S. platensis* proteins. However, the study was limited to in vitro assays; therefore, future in vivo and clinical investigations are required to validate the bioavailability, metabolic stability, and safety of *S. platensis*-derived peptides. Future research should also explore scale-up feasibility, process optimization, and formulation strategies for incorporating *S. platensis* hydrolysates into functional foods, nutraceutical supplements, and antioxidant-enriched formulations.

## Figures and Tables

**Figure 1 foods-14-04242-f001:**
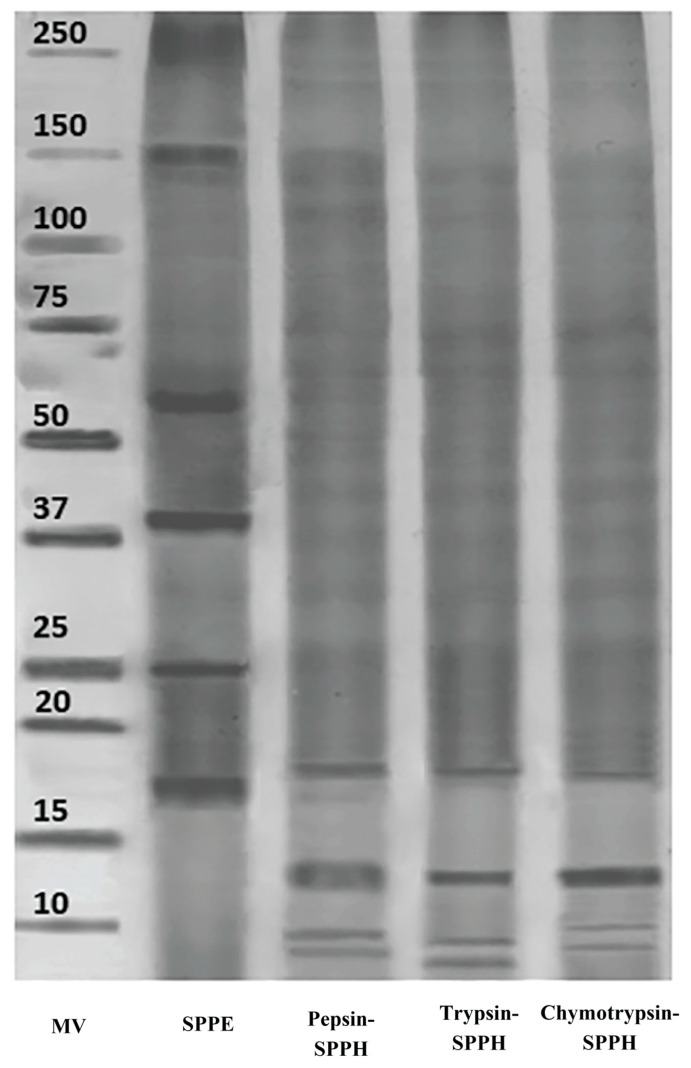
**SDS-PAGE of *S. platensis* protein extract and its derived hydrolysates prepared using pepsin, trypsin, and chymotrypsin individually.** MV: molecular weight, SPPE: *S. platensis* protein extract (5%), SPPH: *S. platensis* protein hydrolysate.

**Figure 2 foods-14-04242-f002:**
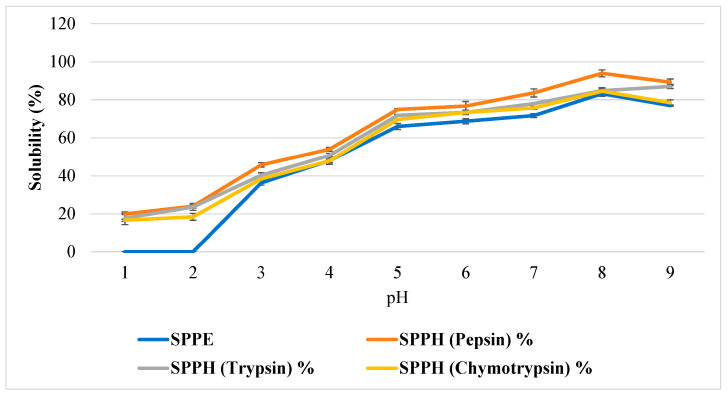
**Solubility of *S. platensis* protein extract (SPPE) and the *S. platensis* protein hydrolysates (SPPHs) formulated using pepsin, trypsin, and chymotrypsin.**

**Table 1 foods-14-04242-t001:** **Enzyme concentration optimization regarding the formation of *S. platensis* protein hydrolysates (SPPHs) for pepsin, trypsin, and chymotrypsin enzymes.**

Pepsin (IU)	Bradford Value Protein Conc. (mg/mL)	Ninhydrin Value Glycine Equivalent (mg/mL)
3% SPPE (Untreated)	22.62 ± 0.03	1.19 ± 0.09
3% SPPH (280)	20.60 ± 0.23	2.54 ± 0.02
3% SPPH (560)	11.56 ± 0.13	3.84 ± 0.06
3% SPPH (1125)	6.09 ± 0.07	4.58 ± 0.08
3% SPPH (2250)	5.67 ± 0.06	6.22 ± 0.05
3% SPPH (4500)	4.20 ± 0.05	7.79 ± 0.05
**Trypsin (IU)**		
3% SPPE (Untreated)	22.62 ± 0.03	1.19 ± 0.09
3% SPPH (250)	20.98 ± 0.13	2.11 ± 0.04
3% SPPH (500)	12.15 ± 0.08	3.34 ± 0.10
3% SPPH (1000)	7.02 ± 0.04	4.54 ± 0.04
3% SPPH (1500)	6.12 ± 0.04	5.48 ± 0.01
3% SPPH (2000)	4.02 ± 0.03	6.14 ± 0.04
**Chymotrypsin (IU)**		
3% SPPE (Untreated)	22.62 ± 0.03	1.19 ± 0.09
3% SPPH (125)	20.75 ± 0.52	2.42 ± 0.11
3% SPPH (250)	17.66 ± 0.44	3.88 ± 0.05
3% SPPH (500)	13.37 ± 0.34	5.16 ± 0.06
3% SPPH (750)	9.92 ± 0.25	6.20 ± 0.08
3% SPPH (1000)	7.01 ± 0.18	6.91 ± 0.08

IU: International Units, SPPE: *S. platensis* protein extract (5%), SPPH: *S. platensis* protein hydrolysate.

**Table 2 foods-14-04242-t002:** **Amino acid composition and protein quality indices of *S. platensis* protein extract and derived hydrolysates prepared from pepsin, trypsin, and chymotrypsin enzymes.**

Amino Acids (g/100 g)	SPPE	Pepsin-SPPH	Trypsin-SPPH	Chymotrypsin-SPPH
Histidine	0.793 ± 0.058	0.989 ± 0.092	0.920 ± 0.087	0.746 ± 0.061
Threonine	2.968 ± 0.068	2.949 ± 0.104	2.956 ± 0.110	2.874 ± 0.034
Methionine + Cysteine	1.144 ± 0.059	0.831 ± 0.060	1.164 ± 0.127	0.971 ± 0.085
Phenylalanine + Tyrosine	4.612 ± 0.045	4.620 ± 0.095	4.560 ± 0.060	4.236 ± 0.054
Leucine	4.031 ± 0.068	4.017 ± 0.092	4.000 ± 0.082	3.880 ± 0.097
Lysine	3.116 ± 0.016	3.073 ± 0.080	2.760 ± 0.044	3.047 ± 0.055
Valine	2.661 ± 0.049	2.707 ± 0.098	2.648 ± 0.059	2.618 ± 0.029
Isoleucine	2.613 ± 0.095	2.505 ± 0.058	2.546 ± 0.069	2.490 ± 0.013
Tryptophan	0.463 ± 0.104	0.231 ± 0.071	0.339 ± 0.042	0.037 ± 0.059
Aspartic acid	4.955 ± 0.020	4.954 ± 0.033	4.984 ± 0.084	4.952 ± 0.068
Glutamic acid	7.955 ± 0.072	7.867 ± 0.089	7.925 ± 0.079	7.883 ± 0.051
Serine	3.038 ± 0.031	2.991 ± 0.104	3.072 ± 0.110	3.043 ± 0.075
Arginine	3.113 ± 0.012	3.050 ± 0.021	2.790 ± 0.065	3.100 ± 0.087
Glycine	2.947 ± 0.046	2.970 ± 0.107	2.896 ± 0.053	2.917 ± 0.044
Alanine	3.375 ± 0.063	3.220 ± 0.007	3.296 ± 0.030	3.318 ± 0.056
**Distribution of amino acids (%)**				
Hydrophobic	28.52	28.19	28.61	28.47
Positive	11.94	12.34	11.18	12.12
Negative	21.96	22.25	22.31	22.57
Polar	20.06	19.59	20.10	19.72
Aromatic	8.63	8.42	8.47	7.51
**Nutritional Parameters**				
PER1	1.14	1.13	1.12	1.07
PER2	1.11	1.11	1.10	1.07
PER3	0.93	0.74	0.94	1.10
%AAS	69	65	62	64
%BV	74.25	71.24	78.83	65.84
%EAA/TAA	46.87	46.66	46.72	45.32

SPPE: *S. platensis* protein extract, SPPH: *S. platensis* protein hydrolysates, PER: protein efficiency ratio, AAS: amino acid score, BV: biological value, EAA: essential amino acids, TAA: total amino acids.

**Table 3 foods-14-04242-t003:** **Antioxidant potential of *S. platensis* protein extract (SPE) and pepsin-, trypsin-, and chymotrypsin-based *S. platensis* protein hydrolysates (SPPHs).**

Sample Type	DPPH IC50 (Trolox µM/mL)	FRAP IC50 (Fe (II) µMol/g)	MDA (µMol/L)
SPPE	0.0092 c ± 0.0061	0.0097 b ± 0.001	0.103 a ± 0.005
Pepsin-SPPH	0.0181 b ± 0.0084	0.0138 a ± 0.0015	0.063 b ± 0.011
Trypsin-SPPH	0.0283 a ± 0.0017	0.0209 a ± 0.004	0.058 b ± 0.012
Chymotrypsin-SPPH	0.0119 bc ± 0.0061	0.0164 a ± 0.0016	0.059 b ± 0.031
*p* value	0.003 *	0.012 *	0.039 *

SPPE: *S. platensis* protein extract (5%), SPPH: *S. platensis* protein hydrolysate. One-way ANOVA with Duncan multiple range test for intergroup differences was applied (level of significance 5%). Heterogeneous letters in a column represent significantly different groups, * Significant difference at *p* value < 0.05.

**Table 4 foods-14-04242-t004:** **Qualitative phytochemicals in *S. platensis* protein extract (SPPE) and pepsin-, trypsin-, and chymotrypsin-based *S. platensis* protein hydrolysates (SPPHs).**

Phytochemical Test	SPPE	Pepsin-SPPH	Trypsin-SPPH	Chymotrypsin-SPPH
Alkaloids	+	+	+	+
Flavonoids	+	+	+	+
Saponin	+	+	+	+
Terpenoids	+	+	+	+
Phytosterol	−	−	−	−
Glycosides	+	−	+	+
Tannin	+	+	−	−
Polyphenols	+	+	+	+

+ Detected, − not detected, SPPE: *S. platensis* protein extract (5%), SPPH: *S. platensis* protein hydrolysate.

**Table 5 foods-14-04242-t005:** **Phytochemical analysis of raw *S. platensis* protein extract and *S. platensis* protein hydrolysates (treated with pepsin, trypsin, and chymotrypsin individually).**

Sample Type	TPC (Gallic Acid µg/mL)	TFC (Catechin mg/mL)	Terpenoids (%)
SPPE	211.28 c ± 3.66	2.24 b ± 0.12	7.63 c ± 0.14
SPPH Pepsin	290.76 b ± 20.48	3.55 a ± 0.33	10.44 a ± 0.18
SPPH trypsin	458.52 a ± 6.10	3.98 a ± 0.23	9.44 ab ± 0.14
SPPH Chymotrypsin	225.59 c ± 0.49	2.12 b ± 0.11	8.60 bc ± 0.13
*p* value	0.000 *	0.001 *	0.026 *

SPPE: *S. platensis* protein extract (5%), SPPH: *S. platensis* protein hydrolysate. One-way ANOVA with Duncan multiple range test for intergroup differences was applied (level of significance 5%). Heterogeneous letters in a column represent significantly different groups, * Significant difference at *p* value < 0.05.

## Data Availability

The raw data supporting the conclusions of this article will be made available by the authors upon request.

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
