# Peer review of "Nutritional, Bio-Functional, and Antioxidant Properties of Enzymatic Hydrolysates Derived from *Spirulina platensis* Proteins"

_foods, 2025, doi:10.3390/foods14244242_

Round 1
Reviewer 1 Report
Comments and Suggestions for Authors
This paper provides important data support for the functionality and bioactivity of spirulina hydrolysates through systematic experimental design and analysis, highlighting its potential applications in antioxidants, nutritional supplementation, and health promotion. It holds certain academic and practical value. However, there are still some areas that need revision, as outlined below:
- In line 11 of the abstract, the term "Spirulina protein extract" and in line 17, the term "Protein Efficiency Ratio" should not have capitalized initials, as they are technical terms. It is recommended to systematically check the capitalization of similar terms throughout the manuscript to ensure consistency in formatting.
- Lines 54-58 in the introduction lack empirical support for the effectiveness of spirulina protein and its hydrolysates. It is suggested to supplement with specific research examples (such as in vitro / in vivo experimental data on certain physiological activities, clinical application cases, etc.). Additionally, the progress in the application of proteases to improve phytochemical components (such as active ingredient release, structural modification, etc.) should be included to enhance the comprehensiveness and persuasiveness of the content.
- Sentences in lines 101-102, 114-115, 147-148, etc., exhibit missing information (e.g., “Bradford assay was performed to assess protein contents in SPPE and the extraction yield, using the method adopted from...”; “The hydrolysis conditions were opted from methods described in and...”). If specific references are available, the complete citation format should be added; if no references can be cited, the detailed operational parameters of the method (such as reagent concentration, reaction temperature, time, etc.) should be provided. It is recommended to review such statements throughout the manuscript and fill in the missing information.
- Lines 109-112 do not clearly state the basis for the setup of different enzyme concentration gradients. Relevant considerations (such as results from preliminary experiments, literature references, theoretical calculations, or practical application needs) should be added.
- Line 127 presents a formatting issue (the specific type is not clarified). The manuscript should refer to the journal's formatting requirements and standardize the use of symbols, paragraph indentation, data presentation, punctuation, and other elements.
- Line 149 does not specify the methods used to determine flavonoids, terpenoids, and total phenolic content. A complete experimental scheme should be provided (such as reagent types, reaction systems, detection wavelengths, standard curve preparation, and other key parameters).
- The Materials and Methods section should specify the sample size corresponding to each detection index (such as the number of samples per group, the number of replicates, etc.) to meet the requirements for experimental reproducibility and statistical analysis.
- Conclusion statements in lines 165-166, 174-175, 194-195, such as “The nutritional composition of spirulina corresponds to most of the available literature, with compositional values residing in a similar range with minute variations.” and “Comparable extraction yields have been reported in and...,” lack supporting evidence. Specific references or comparative data from this study should be added to validate the reliability of the conclusions. It is recommended to review similar statements throughout the manuscript and include necessary experimental data or references.
- An explanation of the mechanisms underlying different hydrolysis results due to different enzymes (such as enzyme source, catalytic specificity, active site, and hydrolysis efficiency) should be added to help readers understand the underlying logic of the experimental results.
- The manuscript mentions an enhancement in the thermal stability of the protein after enzymatic hydrolysis but only links it to protein degradation into shorter peptides. The relationship between peptide chain length, spatial structure of peptide segments, and aggregation state with thermal stability is not fully explained. It is recommended to further elaborate on the molecular mechanisms by which small peptides exhibit superior thermal stability compared to whole proteins (e.g., hydrogen bond network reconstruction, reduced exposure of hydrophobic groups, enhanced anti-aggregation ability).
- The manuscript mentions the effects of phytochemicals (such as phenolics, flavonoids, etc.) on antioxidant activity but does not provide a detailed explanation of the bioactivity and antioxidant mechanisms of these components. For example, how do phenolic and flavonoid compounds specifically enhance antioxidant capacity through mechanisms like free radical scavenging or metal ion chelation? Examples could be added to increase the explanation of the role of phytochemicals in antioxidant activity.
- The conclusion section should clearly point out the innovations, limitations, and suggestions for improvement in this study. It should further clarify the potential applications of spirulina, expand on clinical and industrial research directions, and make the conclusion more comprehensive.
Note: Please insert all references in the corresponding citation locations within the text, ensuring accurate citations that correspond to the main text.
Author Response
|
RESPONSE TO REVIEWER 1 COMMENTS |
||
|
1. Summary |
|
|
|
Thank you very much for taking the time to review this manuscript. Please find detailed responses below and the corresponding revisions highlighted in red in the re-submitted files |
||
|
2. Point-by-point response to Comments and Suggestions for Authors |
||
|
Comments 1: In line 11 of the abstract, the term "Spirulina protein extract" and in line 17, the term "Protein Efficiency Ratio" should not have capitalized initials, as they are technical terms. It is recommended to systematically check the capitalization of similar terms throughout the manuscript to ensure consistency in formatting. |
||
|
Response 1: Thank you for bringing this to our attention. We agree with this comment. Therefore, we have removed the unnecessary capitalization throughout the manuscript |
||
|
Comments 2: Lines 54-58 in the introduction lack empirical support for the effectiveness of spirulina protein and its hydrolysates. It is suggested to supplement with specific research examples (such as in vitro / in vivo experimental data on certain physiological activities, clinical application cases, etc.). Additionally, the progress in the application of proteases to improve phytochemical components (such as active ingredient release, structural modification, etc.) should be included to enhance the comprehensiveness and persuasiveness of the content. |
||
|
Response 2: Thank you for this valuable suggestion. The suggested changes have been made in the introduction to strengthen empirical support regarding the effectiveness of spirulina protein and hydrolysates, focusing on in vitro and in vivo studies. These changes are incorporated in Lines 57-77, along with the addition of new relevant references. |
||
|
Comment 3: Sentences in lines 101-102, 114-115, 147-148, etc., exhibit missing information (e.g., “Bradford assay was performed to assess protein contents in SPPE and the extraction yield, using the method adopted from...”; “The hydrolysis conditions were opted from methods described in and...”). If specific references are available, the complete citation format should be added; if no references can be cited, the detailed operational parameters of the method (such as reagent concentration, reaction temperature, time, etc.) should be provided. It is recommended to review such statements throughout the manuscript and fill in the missing information. |
||
|
Response 3: Thank you for pointing out this referencing mistake. The referred references are added in the manuscript, and all manuscript is revised to ensure the citation of references in accordance with journal's referencing style. |
||
|
2. Comment 4. Lines 109-112 do not clearly state the basis for the setup of different enzyme concentration gradients. Relevant considerations (such as results from preliminary experiments, literature references, theoretical calculations, or practical application needs) should be added. |
||
|
Response 4: We are grateful for this valuable observation; we have added a rationale for using multiple enzyme concentrations for hydrolysis optimization, in light of previous enzyme optimization experiments, along with references, and revised changes are incorporated in Lines 130-134 |
||
|
4. Comment 5: Line 127 presents a formatting issue (the specific type is not clarified). The manuscript should refer to the journal's formatting requirements and standardize the use of symbols, paragraph indentation, data presentation, punctuation, and other elements. |
||
|
Response 5: Thank you for this observation. The formatting issue has been resolved. The missing indentation is added correctly according to the journal’s formatting style. |
||
|
Comment 6: Line 149 does not specify the methods used to determine flavonoids, terpenoids, and total phenolic content. A complete experimental scheme should be provided (such as reagent types, reaction systems, detection wavelengths, standard curve preparation, and other key parameters). |
||
|
Response 6: The research articles from which the exact methodology has been adopted for each experiment have been added to the manuscript. Moreover, a detailed experimental scheme is also added in Lines 203-226 of the methods section. |
||
|
Comment 7: The Materials and Methods section should specify the sample size corresponding to each detection index (such as the number of samples per group, the number of replicates, etc.) to meet the requirements for experimental reproducibility and statistical analysis. |
||
|
Response 7: Thank you for this helpful suggestion. We have revised the methods section to clearly indicate the corresponding sample size and to be in compliance with the statistical requirements. The revised changes can be found in Lines 228-231 |
||
|
Comment 8: Conclusion statements in lines 165-166, 174-175, 194-195, such as “The nutritional composition of spirulina corresponds to most of the available literature, with compositional values residing in a similar range with minute variations,” and “Comparable extraction yields have been reported in and...,” lack supporting evidence. Specific references or comparative data from this study should be added to validate the reliability of the conclusions. It is recommended to review similar statements throughout the manuscript and include necessary experimental data or references. |
||
|
Response 8: We sincerely appreciate this valuable suggestion; specific references have been added in the manuscript. |
||
|
Comment 9: An explanation of the mechanisms underlying different hydrolysis results due to different enzymes (such as enzyme source, catalytic specificity, active site, and hydrolysis efficiency) should be added to help readers understand the underlying logic of the experimental results. |
||
|
Response 9: We appreciate your thoughtful suggestion. The suggested changes have been incorporated and can be found in Lines 273-287. |
||
|
Comment 10: The manuscript mentions an enhancement in the thermal stability of the protein after enzymatic hydrolysis but only links it to protein degradation into shorter peptides. The relationship between peptide chain length, spatial structure of peptide segments, and aggregation state with thermal stability is not fully explained. It is recommended to further elaborate on the molecular mechanisms by which small peptides exhibit superior thermal stability compared to whole proteins (e.g., hydrogen bond network reconstruction, reduced exposure of hydrophobic groups, enhanced anti-aggregation ability). |
||
|
Response 10: Thank you for this valuable comment. The suggested changes have been incorporated into the manuscript and can be found in lines 399-404 |
||
|
Comment 11: The manuscript mentions the effects of phytochemicals (such as phenolics, flavonoids, etc.) on antioxidant activity but does not provide a detailed explanation of the bioactivity and antioxidant mechanisms of these components. For example, how do phenolic and flavonoid compounds specifically enhance antioxidant capacity through mechanisms like free radical scavenging or metal ion chelation? Examples could be added to increase the explanation of the role of phytochemicals in antioxidant activity. |
||
|
Response 11: Thank you for this valuable recommendation. The suggested changes are incorporated in the revised manuscript and can be found in Lines 504-515. |
||
|
Comment 12: The conclusion section should clearly point out the innovations, limitations, and suggestions for improvement in this study. It should further clarify the potential applications of spirulina, expand on clinical and industrial research directions, and make the conclusion more comprehensive. |
||
|
Response 12: The suggested incorporations related to innovations, limitations, and suggestions are being added, and the updated conclusion is added in the revised manuscript, which can be found in Lines 533-543 |
||
|
Note: Please insert all references in the corresponding citation locations within the text, ensuring accurate citations that correspond to the main text. |
||
|
Note Response: Thank you for this observation. All the in-text citations and references are updated in the revised manuscript according to the journal’s guidelines. |
||
|
The English could be improved to more clearly express the research |
||
|
We are grateful for this insightful suggestion. We have thoroughly reviewed the manuscript for all the grammatical mistakes to improve clarity and scientific expression. |
||

Reviewer 2 Report
Comments and Suggestions for Authors
This study investigated the effects of gastrointestinal proteases (pepsin, trypsin, and chymotrypsin) on the hydrolysis of spirulina protein, focusing on its nutritional, functional properties, and antioxidant potential. The research systematically compared the efficacy of these three proteases and correlated the enhanced antioxidant activity with the release of phytochemicals (total phenolics, flavonoids, and terpenoids). The findings clearly demonstrate that the increased phytochemical content following gastrointestinal enzymatic hydrolysis is a key factor contributing to the improved antioxidant capacity. This insight provides a deeper understanding of the mechanisms by which protein hydrolysates gain enhanced bioactivities, offering practical guidance for developing spirulina protein as an ingredient for functional foods and nutraceuticals. However, the following issues should be addressed before publication of this article:
- Lines 10-11: It would be better to specify the species name in the introduction. Please change "Spirulina" to "Spirulina platensis" in the first sentence to maintain consistency with the title and the rest of the manuscript. This will add precision and avoid any potential ambiguity for the reader.
- Lines 41: The term "adjacent side effects" is non-standard and may cause confusion. Please replace it with the standard term "adverse side effects" or, more concisely, "adverse effects" to accurately describe unintended and harmful outcomes.
- Study Design Strategy: The manuscript should provide a clearer description of the experimental replication. Please specify the number of biological and technical replicates performed for each experiment. Additionally, for data presented as mean ± SD, the exact value of 'n' (e.g., the number of independent biological replicates) must be explicitly stated in the figure legends or methods section to allow the reader to assess the robustness of the findings.
- 2.5. Degree of hydrolysis: The formula for Degree of Hydrolysis (DH) appears to contradict its standard definition. As presented, DH (%) = (Total protein - TCA-soluble protein) / Total protein × 100 actually calculates the percentage of unhydrolyzed protein. Could you please clarify if the reported values of 15.6-33.7% indeed represent the non-hydrolyzed fraction rather than the actual DH, and justify the use of this calculation method?
- 2.6. SDS-PAGE: There appears to be a confusion in the reported gel concentrations. The description "resolving (4%) and stacking (12%)" is reversed. As per standard SDS-PAGE protocol, the stacking gel should have a low percentage of polyacrylamide (typically 4-5%) to concentrate the proteins, while the resolving gel should have a higher percentage (e.g., 8-15%) to separate them by molecular weight. Please correct this throughout the manuscript.
- 2.10. Qualitative and quantitative phytochemical analysis: The reference cited for this method is incomplete. Please provide the specific citation number (e.g., [X]) to allow readers to unambiguously identify the source of the protocol.
- Study Limitations: The manuscript should include a discussion on the limitations of the study. For instance, the authors should explicitly address the implications of using a protein extract instead of a purified protein, such as the potential effects of co-existing compounds in the extract on the observed bioactivities.
- Mechanism Discussion: The manuscript attributes the superior performance of the trypsin hydrolysate to its cleavage specificity but does not delve into the underlying mechanism. Could the enhanced bioactivity be because the arginine and lysine residues targeted by trypsin are more prevalent within the "bioactive regions" of spirulina proteins, or is it primarily due to the generation of a peptide size distribution that is optimal for activity?
Author Response
|
RESPONSE TO REVIEWER 2 COMMENTS |
||
|
1. Summary |
|
|
|
Thank you very much for taking the time to review this manuscript. Please find detailed responses below and the corresponding revisions highlighted in red in the re-submitted files |
||
|
|
|
|
|
3. Point-by-point response to Comments and Suggestions for Authors |
||
|
Comment 1: Lines 10-11: It would be better to specify the species name in the introduction. Please change "Spirulina" to "Spirulina platensis" in the first sentence to maintain consistency with the title and the rest of the manuscript. This will add precision and avoid any potential ambiguity for the reader. |
||
|
Response 1: Thank you for pointing out this matter and for your valuable suggestion. The suggested change has been made in the manuscript and can be found in the revised manuscript, highlighted part. |
||
|
|
||
|
Comment 2: Lines 41: The term "adjacent side effects" is non-standard and may cause confusion. Please replace it with the standard term "adverse side effects" or, more concisely, "adverse effects" to accurately describe unintended and harmful outcomes. |
||
|
Response 2: We are grateful for this keen and valuable input. Suggested change has been incorporated into the manuscript as per instructions, and the change can be found in Line 43. |
||
|
|
||
|
Comment 3: Study Design Strategy: The manuscript should provide a clearer description of the experimental replication. Please specify the number of biological and technical replicates performed for each experiment. Additionally, for data presented as mean ± SD, the exact value of 'n' (e.g., the number of independent biological replicates) must be explicitly stated in the figure legends or methods section to allow the reader to assess the robustness of the findings. |
||
|
Response 3: Thank you for this helpful suggestion. We have revised the methods section to clearly indicate the corresponding sample size and to be in compliance with the statistical requirements. The changes revised can be found in Lines 228-230. |
||
|
1. |
||
|
2. Comment 4: 2.5. Degree of hydrolysis: The formula for Degree of Hydrolysis (DH) appears to contradict its standard definition. As presented, DH (%) = (Total protein - TCA-soluble protein) / Total protein × 100 actually calculates the percentage of unhydrolyzed protein. Could you please clarify if the reported values of 15.6-33.7% indeed represent the non-hydrolyzed fraction rather than the actual DH, and justify the use of this calculation method? |
||
|
Response 4: Thank you for this critical observation. The method of TCA soluble nitrogen was used in this particular study, adopted from Hoyle, N. T., & Merritt, J. H. (1994). Quality of fish protein hydrolysates from herring (Clupea harengus). Journal of Food Science, 59(1), 76–79. https://doi.org/10.1111/j.1365-2621.1994.tb06901.x. In which DH can be determined using below below-mentioned equation This method is one of the widely used methods of DH in subsequent hydrolysis research. The formula in the manuscript has been revised to correctly represent the relationship and to ensure consistency with DH reporting. The revised formula can be found in Lines 144-147 along with justification. |
||
|
4. Comment 5: 2.6. SDS-PAGE: There appears to be a confusion in the reported gel concentrations. The description "resolving (4%) and stacking (12%)" is reversed. As per standard SDS-PAGE protocol, the stacking gel should have a low percentage of polyacrylamide (typically 4-5%) to concentrate the proteins, while the resolving gel should have a higher percentage (e.g., 8-15%) to separate them by molecular weight. Please correct this throughout the manuscript. |
||
|
Response 5: We appreciate the careful observation of the reviewer. The concentrations were in reverse order, and now the corrections are being made in the manuscript and can be found in Line 149. |
||
|
|
||
|
Comment 6: 2.10. Qualitative and quantitative phytochemical analysis: The reference cited for this method is incomplete. Please provide the specific citation number (e.g., [X]) to allow readers to unambiguously identify the source of the protocol. |
||
|
Response 6: We are grateful for your valuable input. The missing citation and references are now added in the revised manuscript and can be found in Lines 205-207. |
||
|
|
||
|
Comment 7: Study Limitations: The manuscript should include a discussion on the limitations of the study. For instance, the authors should explicitly address the implications of using a protein extract instead of a purified protein, such as the potential effects of co-existing compounds in the extract on the observed bioactivities. |
||
|
Response 7: Thank you for this critical observation and suggestion. We have acknowledged these suggestions, and the limitation has been discussed in the manuscript for better understanding and interpretation of the findings. The changes can be observed in the revised manuscript and can be found in Lines 489-498, and Lines 516-525 |
||
|
|
||
|
Comment 8: Mechanism Discussion: The manuscript attributes the superior performance of the trypsin hydrolysate to its cleavage specificity but does not delve into the underlying mechanism. Could the enhanced bioactivity be because the arginine and lysine residues targeted by trypsin are more prevalent within the "bioactive regions" of spirulina proteins, or is it primarily due to the generation of a peptide size distribution that is optimal for activity? |
||
|
Response 8: We thank the reviewer for this insightful comment. The Discussion section has been expanded to address the potential mechanisms underlying the superior bioactivity of the trypsin hydrolysate. The changes can be found in Lines 273-287 |
||

Reviewer 3 Report
Comments and Suggestions for Authors
The text has not been revised, and the bibliographical references are missing. The title refers to nutritional value when it only provides data on proteins. It refers to functional value and antioxidants when it should be the same thing. The introduction does not include bibliographical references, so it could be plagiarism. In the materials and methods section, the methods are not described with bibliographical references, and the explanation is insufficient. The results are not discussed with bibliographical references.

Author Response
|
RESPONSE TO REVIEWER 3 COMMENTS |
||
|
1. Summary |
|
|
|
Thank you very much for taking the time to review this manuscript. Please find detailed responses below, and the corresponding revisions highlighted in red in the re-submitted files |
||
|
|
|
|
|
2. Point-by-point response to Comments and Suggestions for Authors |
||
|
Comment 1: The text has not been revised, and the bibliographical references are missing. |
||
|
Response 1: Thank you for this valuable suggestion. All the in-text citations and references have been added and updated according to the requirements of the journal |
||
|
|
||
|
Comment 2: The title refers to nutritional value when it only provides data on proteins. When discussing nutritional value, certain parameters should be determined, such as fatty acids, B vitamins, and minerals, which are so important in this type of product. |
||
|
Response 2: We are grateful for the reviewer’s observation, but the present study evaluates the nutritional value of spirulina proteins, and their hydrolysates have been evaluated using protein-centered indices, i.e., amino acid composition/ distribution, protein efficiency ratio, biological value, and amino acid score. These indices collectively represent the protein quality supporting the study’s focus on protein-based nutritional evaluation. Although we do have the fatty acid composition of whole spirulina platensis (powder), but our main focus was on the investigation of spirulina-based proteins and their hydrolysates. |
||
|
|
||
|
Comment 3 It refers to functional value and antioxidants when it should be the same thing. |
||
|
Response 3 We agree with this valuable suggestion from the reviewer, and the change has been made in the title of the manuscript. |
||
|
|
||
|
2. Comment 4: The introduction does not include bibliographical references, so it could be plagiarism. |
||
|
Response 4: Thank you for this critical observation. All the in-text citations and references have been added and updated according to the requirements of the journal. |
||
|
|
||
|
4. Comment 5: In the materials and methods section, the methods are not described with bibliographical references, and the explanation is insufficient. |
||
|
Response 5: Thank you for this valuable suggestion. Sufficient explanation has been added in the methods section to ensure reproducibility, along with citation of references following journal guidelines |
||
|
|
||
|
Comment 6: The results are not discussed with bibliographical references. |
||
|
Response 6 Results and discussion section have been revised, and all the bibliographical references have been added as per the valuable suggestion. |
||

Round 2
Reviewer 2 Report
Comments and Suggestions for Authors
The auothors have revised the manuscript accordingly.
Author Response
Comment: The authors have revised the manuscript accordingly.
We thank the reviewer for their keen observation and constructive feedback. We appreciate the helpful comments provided throughout the review process
Reviewer 3 Report
Comments and Suggestions for Authors
the corrections are appropriate and the text has gained scientific relevance
Author Response
Comment: the corrections are appropriate and the text has gained scientific relevance
We are extremely grateful to the reviewer for the productive and constructive feedback. We are grateful for this acknowledgment and deeply appreciate the valuable comments received throughout the review process